# Learning to Reinforcement Learn by Imitation

## Abstract

Meta-reinforcement learning aims to learn fast reinforcement learning (RL) procedures that can be applied to new tasks or environments. While learning fast RL procedures holds promise for allowing agents to autonomously learn a diverse range of skills, existing methods for learning efficient RL are impractical for real world settings, as they rely on slow reinforcement learning algorithms for meta-training, even when the learned procedures are fast. In this paper, we propose to learn a fast reinforcement learning procedure through supervised imitation of an expert, such that, after meta-learning, an agent can quickly learn new tasks through trial-and-error. Through our proposed method, we show that it is possible to learn fast RL using demonstrations, rather than relying on slow RL, where expert agents can be trained quickly by using privileged information or off-policy RL methods. Our experimental evaluation on a number of complex simulated robotic domains demonstrates that our method can effectively learn to learn from spare rewards and is significantly more efficient than prior meta reinforcement learning algorithms.

## 1 Introduction

Meta-learning holds the promise of enabling learning systems to compile a diverse set of prior experiences and use this compiled prior knowledge to efficiently learn new skills or rapidly adapt to new environments. Meta reinforcement learning seeks to enable fast learning of new skills through trial-and-error, akin to how humans can rapidly learn how to walk when on ice for the first time, or more quickly learn how to play a new sport given experience with other sports. Hence, meta learning is an important aspect of how humans learn, and is particularly useful in real-world situations with diverse and dynamic environments. Unfortunately, it is challenging to develop meta-reinforcement learning methods that are practical, due to the immense sample complexity of reinforcement learning (RL) methods that is exacerbated by learning-to-learn. For example, prior meta-reinforcement learning methods have reported using more than 250 million transitions for learning to reinforcement learn in tabular MDPs (Duan et al., 2016). Thus, if we want to endow machines with the ability use prior experience to quickly and autonomously adapt to new situations in real world settings, we need to develop more practical algorithms for learning fast reinforcement learning procedures.

We make the following observation in this work: while the goal of meta-reinforcement learning is to acquire fast and efficient reinforcement learning procedures, those procedures themselves do not need to be acquired *through* reinforcement learning. In the same way that humans can be coached (by other humans), a meta-reinforcement learning algorithm can receive more direct supervision during meta-training, in the form of example demonstrations, and then optimize for a purely reward-driven reinforcement learning procedure that converges to solutions that are as good as the provided demonstrations. At meta-test time, when faced with a new task, the method solves the same problem as conventional meta-reinforcement learning: acquiring the new skill using only reward signals. But during meta-training, the availability of demonstrations dramatically improves the efficiency and effectiveness of the method.

Our main contribution is a method that learns fast reinforcement learning via imitation. As illustrated in Figure 1, we optimize for a set of parameters such that only one or a few gradient steps leads to a policy that matches the expert's actions. By using demonstrations during meta-training, there is no challenge with exploration in the meta-optimization, making it possible to effectively learn how to learn in sparse reward environments, including from binary success/failure feedback. While the

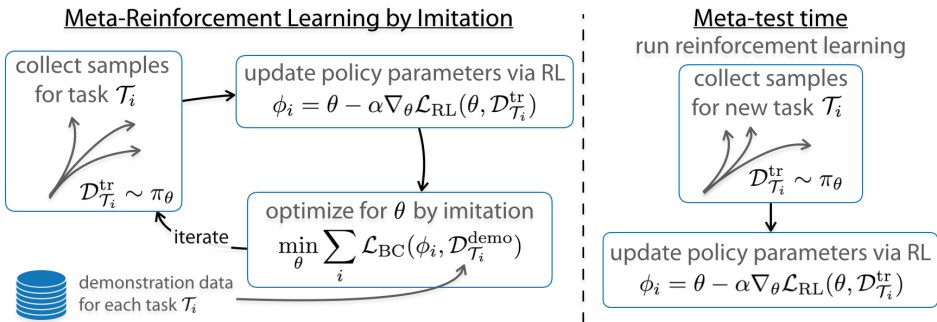

Figure 1: A diagram of our method. We aim to optimize for a fast reinforcement learning procedure through imitation. We train for a set of initial parameters $\theta$ such that only one or a few steps of gradient descent produces a policy that matches the actions from expert demonstrations. At meta-test time, we run a few steps of reinforcement learning to efficiently acquire a policy for a new task.

combination of imitation and RL has been explored before (Peters & Schaal, 2006; Taylor et al., 2011; Brys et al., 2015; Subramanian et al., 2016; Hester et al., 2018; Sun et al., 2018; Rajeswaran et al., 2018; Nair et al., 2018), the particular combination of imitation and RL in a meta-learning context has not been studied in prior work. As we show in our experiments, this combination is in fact extremely powerful: compared to meta-reinforcement learning, our method can meta-learn comparable adaptation skills with up to 10x fewer interaction episodes, making meta-RL much more viable for real-world learning. Further, our experiments indicate that our method can be used to acquire reinforcement learning procedures that effectively learn from sparse rewards in constrained settings.

## 2   RELATED WORK

Our work builds upon prior work on meta-learning (Schmidhuber, 1987; Bengio et al.; Thrun & Pratt, 2012), where the goal is to learn how to efficiently learn. We focus on the particular case where the goal is to learn an efficient reinforcement learner, as typical in the meta-reinforcement learning setting (Schmidhuber, 1987; Duan et al., 2016; Wang et al., 2016; Finn et al., 2017a). Prior works have sought to solve this problem by optimizing for an efficient reinforcement learner through reinforcement learning, where the learner is represented by a recurrent network (Duan et al., 2016; Wang et al., 2016; Mishra et al., 2018; Stadie et al., 2018), gradient descent from a learned initialization (Finn et al., 2017a; Gupta et al., 2018), a learned critic that provides gradients to the policy (Sung et al., 2017; Houthooft et al., 2018), or a planner using an adaptable model (Clavera et al., 2018). The key differentiating factor with this work is that our approach aims to leverage demonstrations for learning an efficient RL procedure. Our algorithm is able to learn from a reasonable number of demonstrations with a very small number of real world trials, making it practical for applications in the physical world, where humans can provide demonstrations efficiently. The demonstrations can also come from an algorithmic expert, yielding an alternative interpretation of our method as a meta-reinforcement learning algorithm that optimizes for expert policies using RL and then separately uses such experts for meta-training.

Our approach is also related to few-shot imitation learning (Duan et al., 2017; Finn et al., 2017b), in that we leverage demonstration data for meta-learning. However, unlike these approaches, we are learning a trial-and-error procedure, and demonstrations are only required during meta-training. Prior work has also sought to learn planning procedures through imitation (Tamar et al., 2016; Karkus et al., 2017; Okada et al., 2017; Srinivas et al., 2018; Choudhury et al., 2018; Lee et al., 2018). We instead learn a trial-and-error RL procedure through imitation.

Meta-learning is closely related to multi-task learning (Caruana, 1998), where the goal is to master a fixed set of predefined tasks (whereas meta-learning seeks to use experience from multiple tasks to quickly master new tasks). In this respect, our approach is related to multi-task learning methods that seek to distill policies for multiple tasks into a single policy, such as guided policy search (Levine et al., 2016), policy distillation (Rusu et al., 2016; Parisotto et al., 2016), and related approaches (Teh et al., 2017; Omidshafiei et al., 2017; Ghosh et al., 2018). Like these prior works, we use a separate expert

for each condition or task, but unlike these approaches, we use these experts to train a meta-learner, rather than a single policy.

Prior methods have also sought to use demonstrations to make standard reinforcement learning more efficient in the single-task setting (Peters & Schaal, 2006; Kober & Peters, 2009; Kormushev et al., 2010; Taylor et al., 2011; Brys et al., 2015; Subramanian et al., 2016; Hester et al., 2018; Sun et al., 2018; Rajeswaran et al., 2018; Nair et al., 2018; Kober et al., 2013; Silver et al., 2016). These prior methods aim to learn a policy from demonstrations and rewards. Our approach instead aims to leverage demonstrations to learn how to efficiently reinforcement learn *without* demonstrations. We use a simple behavioral cloning objective in the outer loop. Ideas from the aforementioned papers, as well as more sophisticated imitation algorithms like DAgger (Ross et al., 2011) and BCO (Torabi et al., 2018), are complementary to our approach and could be used as a drop-in replacement for the outer optimization.

## 3 PRELIMINARIES

As our goal is to learn an RL algorithm in an efficient way, we will build on the model-agnostic meta-learning (MAML) algorithm (Finn et al., 2017a), which has previously been shown to be more data efficient than recurrent meta-learners (Finn et al., 2017b). In this section, we will introduce notation and overview the MAML algorithm.

Meta-learning algorithms optimize for the ability to learn new tasks quickly and efficiently. To do so, they use data collected across a wide range of meta-training tasks and are evaluated based on their ability to learn new meta-test tasks. Meta-learning assumes that the meta-training and meta-test tasks are drawn from some distribution $p(\mathcal{T})$. Generally, meta-learning can be viewed as discovering the structure that exists between tasks such that, when the model is presented with a new task from the meta-test set, it can use the known structure to quickly learn the task. MAML achieves this by optimizing for a deep network's initial parameter setting such that one or a few steps of gradient descent on a few datapoints leads to effective generalization (referred to as few-shot generalization) (Finn et al., 2017a). Then, after meta-training, the learned parameters are fine-tuned on data from a new task.

Concretely, consider a supervised learning problem with a loss function denoted as $\mathcal{L}(\theta, \mathcal{D})$, where $\theta$ denotes the model parameters and $\mathcal{D}$ denotes the labeled data. For a few-shot supervised learning problem, MAML assumes access to a small amount of data for a large number of tasks. During meta-training, a task $\mathcal{T}$ is sampled, along with data from that task, which is randomly partitioned into two sets, $\mathcal{D}^{\text{tr}}$ and $\mathcal{D}^{\text{val}}$. We will assume that $\mathcal{D}^{\text{tr}}$ has $K$ examples. MAML optimizes for a set of model parameters $\theta$ such that one or a few gradient steps on $\mathcal{D}^{\text{tr}}$ produces good performance on $\mathcal{D}^{\text{val}}$. Effectively, MAML optimizes for generalization from $K$ examples. Thus, using $\phi_{\mathcal{T}}$ to denote the updated parameters, the MAML objective is the following:

$$\min_{\theta} \sum_{\mathcal{T}} \mathcal{L}(\theta - \alpha \nabla_{\theta} \mathcal{L}(\theta, \mathcal{D}^{\text{tr}}_{\mathcal{T}}), \mathcal{D}^{\text{val}}_{\mathcal{T}}) = \min_{\theta} \sum_{\mathcal{T}} \mathcal{L}(\phi_{\mathcal{T}}, \mathcal{D}^{\text{val}}_{\mathcal{T}}).$$

where $\alpha$ is a step size that can be set as a hyperparameter or learned. Moving forward, we will refer to the outer objective as the *meta-objective*. Subsequently, at meta-test time, $K$ examples from a new, held-out task $\mathcal{T}_{\text{test}}$ are presented and we can run gradient descent starting from $\theta$ to infer model parameters for the new task:

$$\phi_{\mathcal{T}_{\text{test}}} = \theta - \alpha \nabla_{\theta} \mathcal{L}(\theta, \mathcal{D}^{\text{tr}}_{\mathcal{T}_{\text{test}}}).$$

For convenience, we will use only one inner gradient step in the equations. However, using multiple inner gradient steps is straight-forward, and frequently done in practice.

Finn et al. (2017a) also applied the MAML algorithm to the meta-reinforcement learning setting, where each dataset $\mathcal{D}_{\mathcal{T}_i}$ consists of trajectories of the form $\mathbf{s}_1, \mathbf{a}_1, ..., \mathbf{a}_{H-1}, \mathbf{s}_H$ and where the inner and outer loss function corresponds to the negative expected reward:

$$\mathcal{L}_{\text{RL}}(\phi, \mathcal{D}_{\mathcal{T}_i}) = -\frac{1}{|\mathcal{D}_{\mathcal{T}_i}|} \sum_{\mathbf{s}_t, \mathbf{a}_t \in \mathcal{D}_{\mathcal{T}_i}} r_i(\mathbf{s}_t, \mathbf{a}_t) = -\mathbb{E}_{\mathbf{s}_t, \mathbf{a}_t \sim \pi_{\phi}, q_{\mathcal{T}_i}} \left[ \frac{1}{H} \sum_{t=1}^{H} r_i(\mathbf{s}_t, \mathbf{a}_t) \right]. \quad (1)$$

Here, $q_{\mathcal{T}_i}$ is used to denote the transition dynamics of task $\mathcal{T}_i$. Policy gradients (Williams, 1992) were used to estimate the gradient of this loss function. Thus, the algorithm proceeded as follows: for each

task $\mathcal{T}_i$, first collect samples $\mathcal{D}^{\text{tr}}_{\mathcal{T}_i}$ from the policy $\pi_\theta$, then compute the updated parameters using the policy gradient evaluated on $\mathcal{D}^{\text{tr}}_{\mathcal{T}_i}$, then collect new samples $\mathcal{D}^{\text{val}}_{\mathcal{T}_i}$ via the updated policy parameters, and finally update the initial parameters $\theta$ by taking a gradient step on the meta-objective. In the next section, we will introduce a new approach to meta-reinforcement learning using ideas from the MAML algorithm.

## 4 LEARNING TO REINFORCEMENT LEARN BY IMITATION

In this section, we will start by formalizing the problem setting and assumptions. Then, we will present our approach. We would like our problem definition to encapsulate the setting of learning a reinforcement learner from expert demonstrations. In particular, we will train the agent to be able to quickly reinforcement learn a range of tasks so that, after this meta-training phase, the agent can quickly learn new, related tasks. During meta-training, we will assume that both rewards and demonstrations are available for each of the training tasks, while at meta-test time, only rewards are available for new tasks to be learned. While we assume expert demonstrations during meta-training, these demonstrations could come from a variety of sources including a human expert, an algorithm expert, such as a planner that has access to a simulator, or agent trained via reinforcement learning. Thus, if only rewards are available for each task, as is the case in the meta-reinforcement learning problem, expert demonstrations could be acquired via single-task reinforcement learning, making this problem statement fully compatible with the meta-RL setting.

Like prior meta-learning settings, we will assume a distribution of tasks $p(\mathcal{T})$, where meta-training tasks are drawn from $p$ and meta-testing consists of learning held-out tasks sampled from $p$ using what was learned during meta-training. Each task $\mathcal{T}$ will consist of a finite-horizon Markov decision process with states $\mathbf{s}$, actions $\mathbf{a}$, rewards $r(\mathbf{s}_t, \mathbf{a}_t) \to \mathbb{R}$, unknown dynamics $q(\mathbf{s}_{t+1}|\mathbf{s}_t, \mathbf{a}_t)$, and horizon $H$. Following typical meta-RL approaches, we will assume that the dimensionality of the state and action space is consistent across all tasks in $p(\mathcal{T})$ and that the tasks share common structure that can be used to more efficiently learn new tasks.

For each meta-training task $\mathcal{T}_i$, we assume the ability to observe rewards $r_i$ and access to a set of demonstration trajectories, each denoted as $\mathbf{d}_i := (\mathbf{s}^\star_1, \mathbf{a}^\star_1, ..., \mathbf{s}^\star_H)$, which we assume to be sampled from an optimal policy for task $\mathcal{T}_i$.

### 4.1 META-REINFORCEMENT LEARNING VIA IMITATION

In our algorithm, we aim to learn an initialization of policy parameters such that one or a few policy gradient steps from that initialization leads to effective performance, akin to the MAML algorithm. Because our goal is to learn a reinforcement learning procedure, the inner policy gradient optimization in the MAML algorithm will remain the same. However, the outer objective will now be based on behavior cloning (BC). The essence of the our approach is optimizing the following objective:

$$\min_\theta \sum_{\mathcal{T}_i} \mathcal{L}_{\text{BC}}(\theta - \alpha \nabla_\theta \mathcal{L}_{\text{RL}}(\theta, \mathcal{D}^{\text{tr}}_{\mathcal{T}_i}), \mathcal{D}^{\text{val}}_{\mathcal{T}_i}), \tag{2}$$

where the inner optimization corresponds to gradient descent on $\mathcal{L}_{\text{RL}}$ defined in Equation 1, using policy gradients, and the the outer optimization corresponds to imitation learning using behavior cloning. Since our outer objective corresponds to imitation learning, we choose to have the data $\mathcal{D}^{\text{val}}_{\mathcal{T}_i}$ be the demonstrations for task $\mathcal{T}_i$: $\mathcal{D}^{\text{val}}_{\mathcal{T}_i} := \{\mathbf{d}_{i,j}; j = 1, ..., N\}$. This data is off-policy, as it is collected before meta-training, according to the expert policy for task $\mathcal{T}_i$. As a result, unlike the original MAML RL algorithm, we do not need to collect on-policy data from the adapted policy to evaluate the meta-objective. Concretely, the behavioral cloning loss function is the following:

$$\mathcal{L}_{\text{BC}}(\phi, \mathcal{D}_{\mathcal{T}_i}) \triangleq - \sum_{\mathbf{d}_{i,j} \in \mathcal{D}_{\mathcal{T}_i}} \sum_{\mathbf{s}^\star_t, \mathbf{a}^\star_t \in \mathbf{d}_{i,j}} \log \pi_\phi(\mathbf{a}^\star_t \mid \mathbf{s}^\star_t). \tag{3}$$

Now that we have fully defined our objective, the key question is how to optimize it efficiently. We will generally proceed similar to the MAML algorithm; each meta-iteration consists of the following: for each task $\mathcal{T}_i$, we first draw samples $\mathcal{D}^{\text{tr}}_{\mathcal{T}_i}$ from the policy $\pi_\theta$, then compute the updated policy parameters $\phi_{\mathcal{T}_i}$ using the $\mathcal{D}^{\text{tr}}_{\mathcal{T}_i}$, then we update $\theta$ to optimize $\mathcal{L}_{\text{BC}}$, averaging over all tasks in the minibatch. Note that this requires sampling trajectories from $\pi_\theta$ at every meta-iteration. Thus, for

---

**Algorithm 1** Meta Reinforcement Learning via Imitation (MRI)

**Require:** $p(\mathcal{T})$: distribution over tasks
**Require:** demonstrations $\mathcal{D}_{\mathcal{T}_i} \triangleq \{\mathbf{d}_j\}_i$ for each task $\mathcal{T}_i$
**Require:** $\alpha, \beta$: step size hyperparameters
 1: randomly initialize $\theta$
 2: **while** not done **do**
 3:     Sample task $\mathcal{T}_i \sim p(\mathcal{T})$ {or minibatch of tasks}
 4:     Sample $K$ trajectories $\mathcal{D}^{\text{tr}} = \{(\mathbf{s}_1, \mathbf{a}_1, ...\mathbf{s}_H)\}$ using $\pi_\theta$ in $\mathcal{T}_i$
 5:     $\theta_{\text{init}} \leftarrow \theta$
 6:     **for** $n = 1...N_{\text{BC}}$ **do**
 7:         Evaluate $\nabla_\theta \mathcal{L}_{\text{RL}}(\theta, \mathcal{D}^{\text{tr}})$ according to Eq. 1 and 4 with importance weights $\frac{\pi_\theta(\mathbf{a}_t|\mathbf{s}_t)}{\pi_{\theta_{\text{init}}}(\mathbf{a}_t|\mathbf{s}_t)}$
 8:         Compute adapted parameters with gradient descent: $\phi_i = \theta - \alpha\nabla_\theta \mathcal{L}_{RL}(\theta, \mathcal{D}^{\text{tr}})$
 9:         Sample demonstrations $\mathcal{D}^{\text{val}} \sim \mathcal{D}_{\mathcal{T}_i}$
10:         Update $\theta \leftarrow \theta - \beta\nabla_\theta \mathcal{L}_{\text{BC}}(\phi_i, \mathcal{D}^{\text{val}})$ according to Equation 3
11:     **end for**
12: **end while**

---

minimal on-policy sample requirements, it is crucial to minimize the number of meta-iterations. In particular, while behavior cloning is relatively data efficient, it still often requires a large number of gradient steps. If we only take a single gradient step on the behavioral cloning meta-objective at each meta-iteration, the algorithm will require a substantial number of meta-iterations and, as a result, a substantial number of on-policy samples. To mitigate this, we note that, since behavior cloning is off-policy, we can take *multiple* gradient steps on the meta-objective in each meta-iteration.

Taking many off-policy gradient steps is essential for good sample efficiency; however, doing so is nontrivial. After the first gradient step on the meta-objective modifies the pre-update parameters $\theta$, we need to recompute the adapted parameters $\phi_i$ starting from $\theta$, and we would like to do so *without* collecting new data from $\pi_\theta$. To achieve this, we use an importance-weighted policy gradient, with importance weights $\frac{\pi_\theta(\mathbf{a}_t|\mathbf{s}_t)}{\pi_{\theta_{\text{init}}}(\mathbf{a}_t|\mathbf{s}_t)}$, where $\theta_{\text{init}}$ denotes the policy parameters at the start of the meta-iteration (the parameters under which the data was collected).

Concretely, at the start of a meta-iteration, we sample trajectories $\tau$ from the current policy with parameters denoted as $\theta = \theta_{\text{init}}$. Then, we take many off-policy gradient steps on $\theta$. Each off-policy gradient step involves recomputing the updated parameters $\phi_i$ using importance sampling:

$$\phi_i = \theta + \alpha\mathbb{E}_{\tau\sim\pi_\theta}\left[\frac{\pi_\theta(\tau)}{\pi_{\theta_{\text{init}}}(\tau)}\nabla_\theta \log \pi_\theta(\tau)A_i(\tau)\right] \tag{4}$$

where $A_i$ is the advantage function. Then, the off-policy gradient step is computed and applied using the updated parameters using the behavioral cloning objective in Equation 3:

$$\theta \leftarrow \theta - \beta\nabla_\theta \mathcal{L}_{\text{BC}}(\phi_i, \mathcal{D}_i^{\text{val}}). \tag{5}$$

The entire algorithm is summarized in Algorithm 1 and visualized in Figure 1.

## 4.2 ANALYSIS AS A META-REINFORCEMENT LEARNING ALGORITHM

The meta-learning algorithm proposed in the previous section learns RL procedures from demonstrations. The demonstrations can come from a human expert, when human demonstrations are easy to obtain. The demonstrations can also come from an agent trained on each individual task using reinforcement learning, as we show in our experiments. In this regard, the algorithm can be viewed as a true meta-reinforcement learning algorithm (without any requirement for externally-provided demonstrations), where meta-RL is decomposed into two stages – first learning experts for the individual tasks via RL, and second, learning an agent that can efficiently learn the training tasks through trial-and-error – where the reinforcement learning agent is supervised via the experts acquired in the first stage.

With this decoupling of the outer and inner RL optimization, we can train the experts used for the outer optimization in a variety of ways. For example, we can easily use highly-efficient off-policy reinforcement learning methods, such as actor-critic methods, to train the expert, leading to

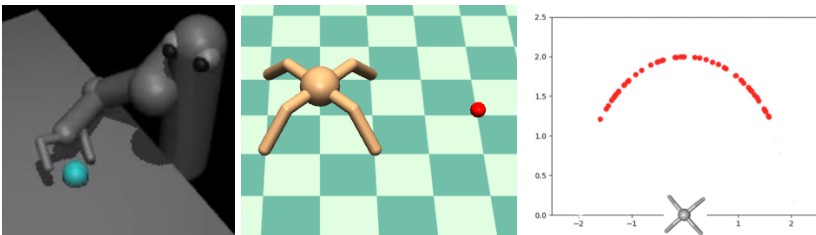

Figure 2: Illustration of a reaching task (left) and an ant locomotion tasks (middle) in our experimental evaluation. The range of ant goals is shown on the right. Our experiments consider a sparse reward variant of each task, where the reward is only provided when the robot moves its gripper next to the cyan ball and when the ant comes near the target location.

significantly improved sample efficiency. In contrast, developing an actor-critic meta-RL algorithm is highly non-trivial, since it involves creating a meta-optimization of a two-player game. Another use-case of decoupling is in learning RL procedures that can learn from sparse rewards. Even if our goal is to learn an RL algorithm that can learn from sparse reward feedback, we can train the experts used for the outer objective using shaped rewards, which is significantly more practical.

In a setting where the expert policy can easily be evaluated on arbitrary states (unlike a human), it is possible to have the outer loop of our algorithm be on-policy, evaluated on the states visited by the updated policy $\pi_{\phi_i}$. From this perspective, our approach is closely related to the multi-task policy distillation methods (Rusu et al., 2016; Parisotto et al., 2016) and guided policy search methods (Levine et al., 2016). Like guided policy search, the RL experts can be trained with extra information, such as better shaped reward functions or additional low-level state information. Further, guided policy search intertwines the two stages so that the experts and learners are trained jointly. Investigating these extensions to our method is an interesting direction for future work.

### 4.3 Algorithm Implementation

To implement the algorithm, we use the TensorFlow library (Abadi et al., 2016). An important aspect of the method is differentiating through the inner policy gradient optimization. While TensorFlow can automatically provide gradients through gradients, it does not take into account the distribution under which the trajectories were sampled and, in particular, the dependence of this distribution on the parameters $\theta$. This dependency is important, as it allows the outer optimization to optimize for initial policies $\pi_\theta$ under which there is sufficient exploration for effective learning. Thus, to implement the correct gradient of the meta-objective, we manually derive the term in the gradient that is missing (see Appendix A for derivation) and add it to the gradient computed with TensorFlow, before passing the corrected gradient to the Adam optimizer. Following the original MAML RL implementation[1], we fit linear baselines independently for each task in the inner loop of the algorithm. Then, these baselines are used to estimate the policy gradient.

## 5 Experimental Evaluation

In designing our evaluation, we aim to answer the following questions: (1) can our method effectively learn an effective reinforcement learning algorithm through imitation?, (2) can our approach learn to efficiently learn from sparse rewards?, (3) does our method improve upon the sample efficiency of meta-RL?

To answer these questions, we will consider three simulated continuous control domains. The first is a point robot navigating in 2D, where different tasks correspond to different goal locations. The second domain involves controlling a 7 DoF arm via torque-control to reach a position on a table, where each task is to different location on the table to reach. For our final, most complex domain, we use a simulated quadruped (or ant), where different tasks correspond to different goal locations in the world. The reaching and ant environments are visualized in Figure 2. We will first consider the setting where the reward functions for the tasks are the negative distance to the goal, which is nicely shaped for reinforcement learning. Then, in Section 5.2, we will consider a sparse-reward setting. We will refer to our approach as Meta-Reinforcement learning via Imitation (MRI).

---

[1]The publicly available MAML RL implementation is at `https://github.com/cbfinn/maml_rl`

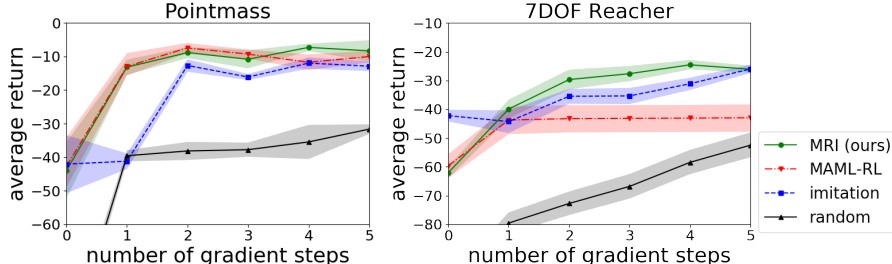

Figure 3: Reinforcement learning performance at meta-test time, comparing our approach with the MAML RL method, learning from a policy initialized from demonstrations and a random initialization. Our method is able to match the performance of meta-reinforcement learning and slightly outperform initializing from demonstrations in these experimental domains

We compare our MRI approach to the original MAML RL method (Finn et al., 2017a), using one step of policy gradient for the inner optimization and trust region policy optimization for the outer optimization. We additionally compare to two baselines. To provide a lower bound on learning performance that indicates the difficulty of the task, we compare to a randomly-initialized policy. We also compare to a policy initialized via behavior cloning on all of the demonstrations. Note that this corresponds to our approach with an inner learning rate of $\alpha = 0$. This comparison measures the effect of explicitly training for fast adaptation, rather than simply training for a policy that attempts to represent all of the meta-training tasks at once.

For all methods, we use policies parameterized by fully-connected networks with two hidden layers of size 100 with ReLU nonlinearities. For both MAML and our method, we use one inner policy gradient step with 20 trajectories and an outer batch size of 40 tasks. For our method, we use $N_{\text{BC}} = 200$ off-policy gradient steps on the imitation objective within each meta-optimization step. For training MRI and the imitation policy, we used 8000 demonstrations for the point robot and reaching domains, drawn collectively from 200 tasks (40 demonstrations per task), and 2000 demonstrations for the ant task, drawn from 100 tasks (20 demonstrations per task).

Videos of our results are available online[2]. To facilitate reproducibility and future work in this direction, we will make the code for our experiments and algorithm available upon publication.

## 5.1 SIMULATED CONTROL RESULTS

We evaluate each method by running reinforcement learning on new tasks drawn from $p(\mathcal{T})$, evaluating the average return after only a small amount of data has been used to adapt the policy to the new task. We report results by plotting learning performance, in terms of average return, as a function of policy gradient steps at meta-test time, where each gradient step uses 20 roll-outs. As seen in Figure 3, we see that our approach is able to achieve comparable performance to the original MAML RL implementation. However our method requires many fewer environment interaction samples during meta-training, because it can take many off-policy updates within a single meta-optimization. On the 7 DoF reaching task, our method required only 5 iterations on the meta-objective to reach good performance, while MAML required more than 50 iterations. Each iteration involves generating 20 sampled trajectories on each of the 40 meta-training tasks, which means that MAML requires 40,000 rollouts during meta-training – a number that would likely be impractical on real physical systems. Our method requires only 4000 roll-outs, well within the realm of feasible training times. The baseline imitation policy also performs surprisingly well on this task, suggesting that a policy that imitates the expert on all tasks provides a good initialization for learning new tasks, at least when using well-shaped reward functions. However, as we show next, such an imitation policy is a less effective initialization when reinforcement learning from sparse rewards.

## 5.2 LEARNING TO LEARN FROM SPARSE REWARDS

One of the potential benefits of learning to learn from demonstrations is that exploration challenges are substantially reduced for the meta-optimizer, compared to using reinforcement learning as the

---

[2]Video results are available at `https://sites.google.com/view/metarl-via-imitation`

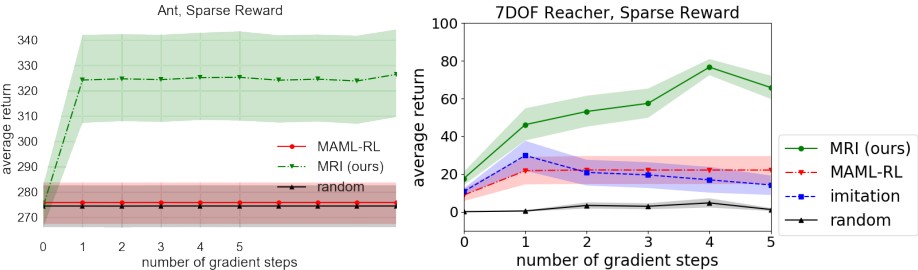

Figure 4: Reinforcement learning performance, when only sparse rewards are available at meta-test time. Our method is able to learn when only sparse rewards are available, whereas prior methods struggle.

meta-optimizer, since the demonstrations provide detailed guidance to indicate to the algorithm how the task should be performed. To test this hypothesis, we experiment with learning to reinforcement learn from sparse reward signals. In our first reaching experiment, the reward is $1$ if a simulated robotic arm moves its gripper inside the target region (see Figure 2) and zero otherwise. In the second locomotion experiment, the reward is $0$ when the ant is far from the goal and a increasing positive value when the ant is close, as it gets closer. Crucially, the position of the goal location is not provided as input to the policy – the meta-learning algorithm must discover an effective strategy for finding the goal through exploration. Learning to effectively learn in such an environment is a compelling application of meta-reinforcement learning, since sparse rewards are particularly challenging for human-designed reinforcement learning algorithms that are not adapted for a particular family of tasks.

We evaluate all methods in both environments, where each approach is provided with 20 roll-outs per gradient step at meta-test time. The results, in Table 4 indicate that, by meta-learning from demonstrations, our approach is able to substantially outperform the alternative methods with sparse rewards. This suggests that the meta-learning process effectively teaches the policy how to explore and discover the sparse high-reward regions, and then rapidly modify the policy to reach them consistently.

## 6    DISCUSSION AND FUTURE WORK

In this work, we presented a meta-reinforcement learning algorithm that learns efficient reinforcement learning procedures via supervised imitation. This enables a substantially more efficient meta-training phase that incorporates expert-provided demonstrations to drastically acceleration the acquisition of reinforcement learning procedures and priors. We believe that our method addresses a major limitation in meta-reinforcement learning: although meta-reinforcement learning algorithms can effectively acquire adaptation procedures that can learn new tasks at meta-test time with just a few samples, they are typically extremely expensive in terms of sample count during meta-training, limiting their applicability to real-world problems. By accelerating meta-training via demonstrations, we can enable sample-efficient learning *both* at meta-training time and meta-test time. Given the efficiency and stability of supervised imitation, we expect our method to be readily applicable to domains with high-dimensional observations, such as images. Further, given the number of samples needed in our experiments, our approach is likely efficient enough to be practical to run on physical robotic systems, learning fast reinforcement learning procedures in the real world. Investigating applications of our approach to real-world reinforcement learning is an exciting direction for future work.

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

## A  DERIVATION OF CORRECTED META-GRADIENT

We will use $\pi_\theta(\tau)$ as shorthand to denote the probability of a trajectory under the policy $\pi_\theta$.

The updated parameters for task $i$ are computed via policy gradient as:

$$\phi_i = \theta + \alpha \mathbb{E}_{\tau \sim \pi_\theta} \left[ \nabla_\theta \log \pi_\theta(\tau) A_i(\tau) \right]$$

where $A$ denotes the advantage.

The outer objective with imitation is the following:

$$\mathcal{L}(\theta) = \sum_i \sum_t \frac{1}{2} ||\pi_{\phi_i}(\mathbf{a}_t|\mathbf{s}_t) - \pi_i^\star(\mathbf{s}_t)||^2$$

Our goal is to minimize the objective with respect to theta by using gradient descent:

$$\theta \leftarrow \theta - \beta \nabla_\theta \mathcal{L}(\theta) = \theta - \beta \nabla_\theta \sum_i \sum_t \frac{1}{2} ||\pi_{\phi_i}(\mathbf{a}_t|\mathbf{s}_t) - \pi_i^\star(\mathbf{s}_t)||^2$$

Here is the gradient of the outer objective:

$$\nabla_\theta \mathcal{L}(\theta) = \sum_i \sum_t \left( \pi_{\phi_i}(\mathbf{a}_t|\mathbf{s}_t) - \pi_i^\star(s_t) \right) \frac{\partial \pi_{\phi_i}}{\partial \phi_i} \frac{\partial \phi_i}{\partial \theta}$$

Consider the last term, $\frac{\partial \phi_i}{\partial \theta}$. The gradient that TensorFlow will compute will ignore the fact that the expectation above is a function of the parameters $\theta$, since this dependency is implicit in the sampled data. Thus, it will derive the following form of $\frac{\partial \phi_i}{\partial \theta}$:

$$\frac{\partial \phi_i}{\partial \theta} := 1 + \alpha \mathbb{E}_{\tau \sim \pi_\theta} \left[ \nabla_\theta^2 \log \pi_\theta(\tau) A(\tau) \right]$$

In actuality, this is missing one term, corresponding to the gradient with respect to the $\theta$ in the expectation. First, let's write out the expectation:

$$\mathbb{E}_{\tau \sim \pi_\theta} \left[ \nabla_\theta \log \pi_\theta(\tau) A_i(\tau) \right] = \int p_\theta(\tau) \nabla_\theta \log \pi_\theta(\tau) A_i(\tau) d\tau$$

With the above, we can derive the correct form of $\frac{\partial \phi_i}{\partial \theta}$ :

$$\frac{\partial \phi_i}{\partial \theta} := 1 + \alpha \int \nabla_\theta \pi_\theta(\tau) \nabla_\theta \log \pi_\theta(\tau) A(\tau) d\tau + \alpha \mathbb{E}_{\tau \sim \pi_\theta} \left[ \nabla_\theta^2 \log \pi_\theta(\tau) A(\tau) \right] \tag{6}$$

$$= 1 + \alpha \int \pi_\theta(\tau) \nabla_\theta \log \pi_\theta(\tau) (\nabla_\theta \log \pi_\theta(\tau))^T A(\tau) d\tau + \alpha \mathbb{E}_{\tau \sim \pi_\theta} \left[ \nabla_\theta^2 \log \pi_\theta(\tau) A(\tau) \right] \tag{7}$$

$$= 1 + \alpha \mathbb{E}_{\tau \sim \pi_\theta} \left[ \nabla_\theta \log \pi_\theta(\tau) (\nabla_\theta \log \pi_\theta(\tau))^T A(\tau) \right] + \alpha \mathbb{E}_{\tau \sim \pi_\theta} \left[ \nabla_\theta^2 \log \pi_\theta(\tau) A(\tau) \right] \tag{8}$$

$$\frac{\partial \phi_i}{\partial \theta} := 1 + \alpha \int \nabla_\theta \pi_\theta(\tau) \nabla_\theta \log \pi_\theta(\tau) A(\tau) d\tau + \alpha \mathbb{E}_{\tau \sim \pi_\theta} \left[ \nabla_\theta^2 \log \pi_\theta(\tau) A(\tau) \right] \tag{9}$$

$$= 1 + \alpha \int \pi_\theta(\tau) \nabla_\theta \log \pi_\theta(\tau) (\nabla_\theta \log \pi_\theta(\tau) A(\tau))^T d\tau + \alpha \mathbb{E}_{\tau \sim \pi_\theta} \left[ \nabla_\theta^2 \log \pi_\theta(\tau) A(\tau) \right] \tag{10}$$

$$= 1 + \alpha \mathbb{E}_{\tau \sim \pi_\theta} \left[ \nabla_\theta \log \pi_\theta(\tau) (\nabla_\theta \log \pi_\theta(\tau) A(\tau))^T \right] + \alpha \mathbb{E}_{\tau \sim \pi_\theta} \left[ \nabla_\theta^2 \log \pi_\theta(\tau) A(\tau) \right] \tag{11}$$

Therefore, we need to correct the gradient of the meta-objective from TensorFlow by adding the following correction $\mathbf{c}$:

$$\mathbf{c} = \sum_i \sum_t \left( \pi_{\phi_i}(a_t|s_t) - \pi_i^\star(s_t) \right) \frac{\partial \pi_{\phi_i}}{\partial \phi_i} \alpha \mathbb{E}_{\tau \sim \pi_\theta} \left[ \nabla_\theta \log \pi_\theta(\tau) (\nabla_\theta \log \pi_\theta(\tau))^T A_i(\tau) \right]$$

$$\mathbf{c} = \sum_i \sum_t \left( \pi_{\phi_i}(a_t|s_t) - \pi_i^\star(s_t) \right) \frac{\partial \pi_{\phi_i}}{\partial \phi_i} \alpha \mathbb{E}_{\tau \sim \pi_\theta} \left[ \nabla_\theta \log \pi_\theta(\tau)(\nabla_\theta \log \pi_\theta(\tau) A_i(\tau))^T \right]$$

We estimate this with

$$\hat{\mathbf{c}} = \sum_i \sum_t \left( \pi_{\phi_i}(a_t|s_t) - \pi_i^\star(s_t) \right) \frac{\partial \pi_{\phi_i}}{\partial \phi_i} \alpha \left\langle \left( \nabla_\theta \log \pi_\theta(\tau) \right) \left( \nabla_\theta \log \pi_\theta(\tau) A_i(s_{t,p}) \right)^T \right\rangle_\tau$$

where the average is taken over all sampled trajectories $\tau$ for task $i$.

Since $\left\langle \left( \nabla_\theta \log \pi_\theta(\tau) \right) \left( \nabla_\theta \log \pi_\theta(\tau) A_i(s_{t,p}) \right)^T \right\rangle_\tau$ is a square matrix with dimensions corresponding to the number of timesteps per trajectory, we use the following transformation to speed up the calculation of $\hat{\mathbf{c}}$ in TensorFlow. We set

$$F(\theta, \rho, \xi) = \sum_i \sum_t \left( \pi_{\phi_i}(a_t|s_t) - \pi_i^\star(s_t) \right) \frac{\partial \pi_{\phi_i}}{\partial \phi_i} \alpha \left\langle \left( \log \pi_\xi(\tau) \right) \left( \log \pi_\rho(\tau) A_i(s_{t,p}) \right)^T \right\rangle_\tau$$

and calculate

$$\hat{\mathbf{c}} = \nabla_\rho \nabla_\xi F|_{\rho=\xi=\theta}$$

.

Note that, this can be equivalently written as follows:

$$\mathbf{c} = \sum_i \sum_t \left( \pi_{\phi_i}(\mathbf{a}_t|\mathbf{s}_t) - \pi_i^\star(\mathbf{s}_t) \right) \alpha \mathbb{E}_{\tau \sim \pi_\theta} \left[ \left( \frac{\partial \pi_{\phi_i}}{\partial \phi_i} \nabla_\theta \log \pi_\theta(\tau) \right) \left( \nabla_\theta \log \pi_\theta(\tau) \right)^T A_i(\tau) \right]$$

by moving the gradient inside the expectation. This approach will be more computationally efficient, to first compute the inner product $\left( \frac{\partial \pi_{\phi_i}}{\partial \phi_i} \nabla_\theta \log \pi_\theta(\tau) \right)$, which avoids computing a full $N \times N$ matrix outer product.

To implement this corrected meta-gradient, we first get the the incorrect gradient from TensorFlow, derived above. Then, we compute the correction term by getting the two individual gradient terms in the correction $\mathbf{c}$ and computing $\mathbf{c}$ manually. Finally, we add together the gradient and the correction term and pass the resulting, correct gradient to Adam, using the `apply_gradients` function.

