# OpenReview forum: "Learning to Reinforcement Learn by Imitation"
_ICLR.cc/2019/Conference_

### Official Review · AnonReviewer3 · 2018-11-03
**Lack of clarity in core sections**

**Rating:** 5
**Confidence:** 2

**Review:**

This paper describes a meta-RL algorithm through imitation on RL policies. While the paper builds nicely up to the core part, I find essential details missing about the imitation setup. By glancing at previous BC papers (some of which are cited), the quantity for supervised imitations, etc., were clearly defined.

It will be useful for this reviewer if the authors can provide more clarity in explaining the BC task involved in their algorithm.

---

### Official Review · AnonReviewer1 · 2018-11-07
**Very minor contribution, a manuscript that is lacking important details and does not relate it's technical section to existing work, with very thin evaluation.**

**Rating:** 2
**Confidence:** 5

**Review:**

This work addresses the problem of learning a policy-learning-procedure, through meta-learning, that can adapt quickly to new tasks. This work uses MAML for meta-learning, and with this choice, the problem can be broken down into two loops:

1) inner loop: adapting a policy \pi_phi based on unseen rollouts, where initial parameters phi were provided by the meta-trainer in the outer loop
2) outer loop: the meta-trainer tries to learn parameters phi on batches of tasks that provide good initial parameters

In prior work on meta-reinforcement learning via MAML, both the outer as well as inner objective attempt to minimize a RL objective, leading to an algorithm that has very high sample-complexity. This work uses imitation learning for the outer loop procedure, to significantly decrease sample-complexity.

Technical Contribution:
-----------------------
The idea of using imitation learning for reinforcement learning is well explored in the literature, and so using this idea in itself is not real contribution. There are several issues with the presentation of this work, that make it incredibly difficult to identify a technical contribution:

1. overreaching statements without details to backup: you are writing the paper as if you are learning a "RL algorithm" that can be used to quickly learn new tasks. your manuscript does not really provide a description for this "algorithm". After re-reading several other papers I concluded that what you mean is that you learn an initial set of policy parameters that can quickly adapt to new related tasks and an update rule with which you update these parameters. However, standard MAML uses SGD as an update rule so there is really nothing to be learned here. Unfortunately, your paper provides zero detail on these claims of learning a "RL procedure", so for now I have to assume that you are simply learning a good initial set of policy parameters through meta-learning. If that is the case, then using imitation learning in this setting is really not novel, this has been done by a lot of other people before (you're just using MAML to learn "better" initial parameters).
2. you're technical section (section 4) provides some details on the technical challenges of using demonstrations to perform the outer loop optimization step. Unfortunately, you are not putting your work in the context of existing work ([1], [2]), that discuss and address the importance/issue of sampling in meta-rl with MAML. So it's impossible to know whether there is any new insight here

Experimental Evaluation:
-------------------------
The experimental evaluation is very "thin", other than the original MAML-RL and pure imitation learning no other more recent baselines ([1], [2]) have been compared to. And only 2 relatively simple simulation settings are tested.

Summary:
-----------
Very minor contribution, a manuscript that is lacking important details and does not relate it's technical section to existing work, with very thin evaluation.


[1] The Importance of Sampling in Meta-Reinforcement Learning, NIPS 2018
[2] CONTINUOUS ADAPTATION VIA META-LEARNING IN NONSTATIONARY AND COMPETITIVE ENVIRONMENTS, ICLR 2018

---

### Official Review · AnonReviewer5 · 2018-11-18
**Lack of details and reproducibility**

**Rating:** 3
**Confidence:** 2

**Review:**

The paper presents a meta-RL method extends previous work on meta-RL by including an imitation learning step. It is mentioned that the behaviour closing part of this extended algorithm can come from a teacher or some other source. Since this extension is the major contribution, it must be discussed in more detail. I also don't understand why

My second problem with the paper is reproducibility. The purpose of OpenAI is comparability and reproducibility of algorithms. It is not sufficient to simply state that you have used TensorFlow. We need information about the architecture, etc so that the results can be reproduced. Also, since you use OpenAI, the score for each experiment should be compared to the best scores known from the website so that the performance of the new algorithm can be compared to others.

---

### Official Review · AnonReviewer4 · 2018-11-21
**Review of "Learning to Reinforcement Learning by Imitation"**

**Rating:** 4
**Confidence:** 3

**Review:**

This paper proposes a meta-learning algorithm for reinforcement learning that incorporates expert demonstrations. The goal is to reduce the sample complexity of meta-RL algorithms in the validation phase. The paper provides a good discussion of the background literature. Experimental results are provided on multi-goal planning problems for three prototypical simulated systems, namely, a 2D point-mass robot, a 7-DOF manipulator and a quadruped crawler.

The theoretical and practical contributions of this paper are minor. The authors propose a straight-forward combination of MAML, importance-weighted policy gradients across the inner-outer loop and off-policy supervised learning of the expert demonstrations, all standard techniques in reinforcement learning and meta-learning. The experimental section is unconvincing and lacking in details. I however find the approach well-motivated and pertinent. Demonstrations on a real robotic platform, where the improved sample complexity is essential, would make this paper much more impressive.

Detailed comments:

1. Can you compare the different algorithms using the number of rollouts as the X-axis? The narrative provides this information but it is difficult to judge the performance based on Figs. 2 and 3.
2. It is unfair to compare MRI (the approach of this paper) with MAML which does not have access to expert demonstrations for validation tasks. The improved sample complexity is thus directly coming from demonstrations. It is difficult to compare MRI and MAML-RL/Imitation on an equal footing. Perhaps the validation tasks could be significantly harder, e.g., sub-goals in the planning problems, or one could consider a large number of validation tasks.
3. It seems the improvement over MAML-RL/Imitation in Fig. 3 is minor. Why is this so?

---

### Meta-Review · Area_Chair1 · 2018-12-18
**Lacking important details, relation to previous work, experiments**

**Confidence:** 4
**Recommendation:** Reject

**Metareview:**

This paper proposes a meta-learning algorithm for reinforcement learning that incorporates expert demonstrations. The objective is to improve sample efficiency, which is an important problem.

The referees find the approach well-motivated and pertinent, but the theoretical and practical contributions of the paper too slim. A concern was also raised in regard to reproducibility of the results, missing details about the implementation and comparisons with previous results.

The authors did not respond to the reviews.

The four referees are not convinced by this paper, with ratings from strong reject to ok, but not good enough.